# COUNTERFACTUAL FAIRNESS THROUGH DATA PREPROCESSING

## ABSTRACT

Machine learning has become more important in real-life decision-making but people are concerned about the ethical problems it may bring when used improperly. Recent work brings the discussion of machine learning fairness into the causal framework and elaborates on the concept of Counterfactual Fairness. In this paper, we develop the Fair Learning through dAta Preprocessing (FLAP) algorithm to learn counterfactually fair decisions from biased training data and formalize the conditions where different data preprocessing procedures should be used to guarantee counterfactual fairness. We also show that Counterfactual Fairness is equivalent to the conditional independence of the decisions and the sensitive attributes given the processed non-sensitive attributes, which enables us to detect discrimination in the original decision using the processed data. The performance of our algorithm is illustrated using simulated data and real-world applications.

## 1 INTRODUCTION

The rapid popularization of machine learning methods and the growing availability of personal data have enabled decision-makers from various fields such as graduate admission (Waters & Miikkulainen, 2014), hiring (Ajunwa et al., 2016), credit scoring (Thomas, 2009), and criminal justice (Brennan et al., 2009) to make data-driven decisions efficiently. However, the community and the authorities have also raised concern that these automatically learned decisions may inherit the historical bias and discrimination from the training data and would cause serious ethical problems when used in practice (Nature Editorial, 2016; Angwin & Larson, 2016; Dwoskin, 2015; Executive Office of the President et al., 2016).

Consider a training dataset $\mathcal{D}$ consisting of sensitive attributes $S$ such as gender and race, non-sensitive attributes $A$ and decisions $Y$. If the historical decisions $Y$ are not fair across the sensitive groups, a powerful machine learning algorithm will capture this pattern of bias and yield learned decisions $\hat{Y}$ that mimic the preference of the historical decision-maker, and it is often the case that the more discriminative an algorithm is, the more discriminatory it might be.

While researchers agree that methods should be developed to learn fair decisions, opinions vary on the quantitative definition of fairness. In general, researchers use either the *observational* or *counterfactual* approaches to formalize the concept of fairness. The observational approaches often describe fairness with metrics of the observable data and predicted decisions (Hardt et al., 2016; Chouldechova, 2017; Yeom & Tschantz, 2018). For example, Demographic Parity (DP) or Group Fairness (Zemel et al., 2013; Khademi et al., 2019) considers the learned decision $\hat{Y}$ to be fair if it has the same distribution for different sensitive groups, i.e., $P(\hat{Y}|S = s) = P(\hat{Y}|S = s')$. The Individual Fairness (IF) definition (Dwork et al., 2012) views fairness as treating similar individuals similarly, which means the distance between $\hat{Y}(s_i, a_i)$ and $\hat{Y}(s_j, a_j)$ should be small if individuals $i$ and $j$ are similar.

The other branch of fairness and/or discrimination definitions are built upon the causal framework of Pearl (2009a), such as direct/indirect discrimination (Zhang et al., 2017; Nabi & Shpitser, 2018), path-specific effect (Wu et al., 2019b), counterfactual error rate (Zhang & Bareinboim, 2018a) and counterfactual fairness (Kusner et al., 2017; Wang et al., 2019; Wu et al., 2019a). These definitions often involve the notion of counterfactuals, which means what the attributes or decision would be

if an individual were in a different sensitive group. With the help of the potential outcome concept, the measuring of fairness is no longer restricted to the observable quantities (Kilbertus et al., 2017; Zhang & Bareinboim, 2018b). For example, the Equal Opportunity (EO) definition Wang et al. (2019) has the same idea as IF but it can directly compare the actual and counterfactual decisions of the same individual instead of the actual decisions of two similar individuals. The Counterfactual Fairness (CF) definition (Kusner et al., 2017) or equivalently, the Affirmative Action (AA) definition (Wang et al., 2019) goes one step further than EO and derives the counterfactual decisions from the counterfactual non-sensitive attributes. We adopt CF as our definition of fairness and it is formally described in Section 2. We believe causal reasoning is the key to fair decisions as DeDeo (2014) pointed out that even the most successful algorithms would fail to make fair judgments due to the lack of causal reasoning ability.

For the observational definitions, fair decisions can be learned by solving optimization problems, either adding the fairness condition as a constraint (Dwork et al., 2012) or directly optimize the fairness metric as an object (Zemel et al., 2013). When using the counterfactual definitions, however, an approximation of the causal model or the counterfactuals is often needed since the counterfactuals are unobservable. In the FairLearning algorithm proposed by Kusner et al. (2017), the unobserved parts of the graphical causal model are sampled using the Markov chain Monte Carlo method. Then they use only the non-descendants of $S$ to learn the decision, which ensures CF but will have a low prediction accuracy. In Wang et al. (2019), the counterfactual of $A$ had $S$ been $s'$ is imputed as the sum of the counterfactual group mean $\mathbb{E}(A|S = s')$ and the residuals from the original group $A - \mathbb{E}(A|S = s)$. As we discuss later, this approach would only work when a strong assumption of the relationship between $A$ and $S$ is satisfied.

## 1.1 Contributions

We develop the Fair Learning through dAta Preprocessing (FLAP) algorithm to learn counterfactually fair decisions from biased training data. While current literature is vague about the assumptions needed for their algorithms to achieve fairness, we formalize the weak and strong conditions where different data preprocessing procedures should be used to guarantee CF and prove the results under the causal framework of Pearl (2009a). We show that our algorithm can predict fairer decisions with similar accuracy when compared with other counterfactual fair learning algorithms using three simulated datasets and three real-world applications, including the loan approval data from a fintech company, the adult income data, and the COMPAS recidivism data.

On the other hand, the processed data also enable us to detect discrimination in the original decision. We prove that CF is equivalent to the conditional independence of the decisions and the sensitive attributes given the processed non-sensitive attributes under certain conditions. Therefore any well-established conditional independence tests can be used to test CF with the processed data. To our knowledge, it is the first time that a formal statistical test for CF is proposed. We illustrate the idea using the Conditional Distance Correlation test (Wang et al., 2015) in our simulation and test the fairness of the decisions in the loan approval data using a parametric test.

## 2 Causal Model and Counterfactual Fairness

For the discussion below, we consider the sensitive attributes $S \in \mathcal{S}$ to be categorical, which is a reasonable restriction for the commonly discussed sensitive information such as race and gender. The non-sensitive attributes $A \in \mathcal{A} \subseteq \mathbb{R}^d$, and the decision $Y$ is binary as admit or not in graduate admission, hire or not in the hiring process, approve or not in loan assessment.

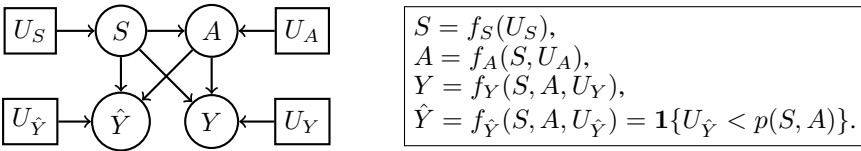

$$S = f_S(U_S),$$
$$A = f_A(S, U_A),$$
$$Y = f_Y(S, A, U_Y),$$
$$\hat{Y} = f_{\hat{Y}}(S, A, U_{\hat{Y}}) = \mathbf{1}\{U_{\hat{Y}} < p(S, A)\}.$$

Figure 1: Structural causal model.

To bring the discussion of fairness into the framework of causal inference, we begin by constructing the Structural Causal Model (SCM) for the data. As described in Pearl (2009b), an SCM $M$ consists of a set of exogenous variables $U$, a set of endogenous variables $V$, and $F$, a set of functions that assign value to each endogenous variable given its parents in $V$ and the exogenous variables $U$. In our case (Figure 1), we consider $V = \{S, A, Y, \hat{Y}\}$, where $\{S, A, Y\}$ are the observed data and $\hat{Y}$ is the prediction of $Y$ we made based on $S$ and $A$. The only exogenous variable affecting $\hat{Y}$ is a $\text{Uniform}(0, 1)$ random variable $U_{\hat{Y}}$ so that we can conveniently express the value of $\hat{Y}$ with a structural equation. We assume that $U_S$, $U_A$, and $U_Y$, which are the exogenous variables that affect $S$, $A$, and $Y$ respectively, are independent of each other. The structural equations on the right side of Figure 1 are described with the functions in $F$, one for each component in $V$. Here we express $f_{\hat{Y}}$ as an indicator function so that $\hat{Y}$ is a Bernoulli random variable that takes value one with probability $p(S, A)$. In general, $p(s, a)$ could be any function that maps $\mathcal{S} \times \mathcal{A}$ to $[0, 1]$, but we are more interested in such functions that will result in a fair decision, more details of which will be discussed in Section 3. It can be seen that the subset of exogenous variables $\{U_S, U_A, U_Y\}$ characterize everything we should know about a unit. Any two units with the same realization will have the same behavior and result irrespective of the other differences in their identities.

Here we give a simplified loan approval model as a running example to help understand the SCM we considered.

**Example 1.** *A bank asks each loan applicant for her/his race $S$ and annual income $A$ to decide to approve the application ($Y = 1$) or not ($Y = 0$). There are two races in the population of the applicants, $S = 1$ represents the advantageous group, and $S = 0$ for the disadvantageous one. Let $U_S \sim \text{Uniform}(0, 1)$, we generate $S = \mathbf{1}\{U_S < 0.7\}$. The annual income is log-normally distributed for each race group and its scale and location parameters may depend on race:*

$$A = c_1 \exp\{c_2 + \lambda_a S + c_3 \sigma_a^S U_A\},$$

*where $U_A$ is a standard normal random variable, $c_1, c_3 > 0$, and $c_2$ are constants that affect the median and spread of the population income, $\lambda_a$ decides the difference in mean log income between the two race groups, and $\sigma_a > 0$ determines the standard deviation ratio of the log incomes. The decision by the bank can be simulated from a logistic model:*

$$Y = \mathbf{1}\{U_Y < \text{expit}(\beta_0 + \beta_a A + \beta_s S)\},$$

*where $U_Y \sim \text{Uniform}(0, 1)$ and $\text{expit}(u) = (1 + e^{-u})^{-1}$.*

In this example, $\beta_s$ characterizes the direct effect of the sensitive attribute on the decision: when $\beta_s > 0$, the applications from the advantageous group are more likely to be approved by the bank when holding the income fixed. On the other hand, $\lambda_a$ partly describes the indirect effect because when both $\lambda_a$ and $\beta_a$ are positive, the advantageous group will have a higher income than the other group on average and thus be favored by the bank even if $\beta_s = 0$. It is worth noting that, apart from the difference in the mean, the difference in higher moments could also cause unfairness indirectly as alluded to in Fuster et al. (2018). In general, if there are any differences in the distribution of $A$ across the categories in $\mathcal{S}$, the decision based on $A$ might be unfair. However, the indirect effect caused by the differences in the higher moments of $A$ could be case dependent and thus harder to interpret. In our case, $\sigma_a > 1$ will lead to a higher average income and hence higher approval probability on average for the advantageous group since the income distribution is right-skewed.

With the SCM in hand, we are ready to define the causal quantity we are interested in. Since most sensitive attributes, such as gender and race, cannot be altered in experiments, we will look into the counterfactuals, namely, what the results $Y$ would be had $S$ been different from the observed facts. This quantity is expressed as $Y_s(U)$ had $S$ been $s$ for a random unit with exogenous variables $U$ sampled from the population. Define $M_s$ to be the modified SCM from $M$ (Figure 1) with the equation for $S$ replaced with $S = s$. Then for any realization $U = u$, the unit level counterfactuals $Y_s(u)$ can be calculated from $M_s$. Similarly, we can define $\hat{Y}_s(U)$ and $\hat{Y}_s(u)$ as the counterfactual predicted decision and its realization. The counterfactual fairness can then be defined on both the decision and the prediction based on the counterfactual result. Here we denote $\mathcal{Y}$ as a placeholder for either $Y$ or $\hat{Y}$.

**Definition 1.** *Counterfactual Fairness. Given a new pair of attributes $(s^*, a^*)$, a (predicted) decision $\mathcal{Y}$ is counterfactually fair if for any $s' \in \mathcal{S}$,*

$$\mathcal{Y}_{s'}(U)|\{S = s^*, A = a^*\} \stackrel{d}{=} \mathcal{Y}_{s^*}(U)|\{S = s^*, A = a^*\}.$$

In other words, the conditional distribution of the counterfactual result should not depend on the sensitive attributes. It should be noted that there are two stages in evaluating the conditional counterfactuals. The first is updating the conditional distribution of $U$. Take the decision $Y$ from Example 1, if $s^* = 0$, then $U_S|\{S = s^*, A = a^*\}$ is from $\text{Uniform}(0.7, 1)$ and $U_A|\{S = s^*, A = a^*\}$ is a constant $(\log(a^*/c_1) - c_2)/c_3$, but $U_Y|\{S = s^*, A = a^*\}$ is still a $\text{Uniform}(0,1)$ random variable since $U_Y$ is independent of $S$ and $A$ from the SCM. The next stage is deriving the conditional distribution of the counterfactuals from the structural equations of $M_s$ and the conditional distribution of $U$. Continuing with our example, $Y_1(U)|\{S = 0, A = a^*\}$ would be equal in distribution to

$$f_Y(1, f_A(1, U_A), U_Y)|\{S = 0, A = a^*\}$$
$$\stackrel{d}{=} f_Y(1, f_A(1, (\log(a^*/c_1) - c_2)/c_3), U_Y)$$
$$\stackrel{d}{=} \mathbf{1}\{U_Y < \text{expit}(\beta_0 + \beta_a c_1 (a^*/c_1)^{\sigma_a} \exp\{\lambda_a + (1 - \sigma_a)c_2\} + \beta_s)\}$$

and $Y_0(U)|\{S = 0, A = a^*\} \stackrel{d}{=} \mathbf{1}\{U_Y < \text{expit}(\beta_0 + \beta_a a^*)\}$. Thus the bank's decision $Y$ would be counterfactually fair if $\sigma_a = 1$, $\lambda_a = 0$ and $\beta_s = 0$.

## 3 PREPROCESSING, LEARNING, AND TESTING

Define a preprocessing procedure $\mathcal{P}^{\mathcal{D}}(s, a) : \mathcal{S} \times \mathcal{A} \to \mathcal{A}'$ to be a function that maps attributes $(s, a)$ to the processed attributes $a'$ given the training data $\mathcal{D}$. Here we consider two such procedures. Denote $\mathbb{P}_n(S = s)$ as the empirical p.m.f. of $S$ and $\mathbb{E}_n(A|S = s)$ as the empirical conditional mean of $A$ given $S$ learned from data $\mathcal{D}$.

**Definition 2** (Orthogonalization). *An orthogonalization procedure $\mathcal{P}_O^{\mathcal{D}}$ is a preprocessing procedure such that*

$$\mathcal{P}_O^{\mathcal{D}}(s^*, a^*) = \sum_s \hat{a}(s)\mathbb{P}_n(S = s),$$

*where $\hat{a}(s) = a^* - \mathbb{E}_n(A|S = s^*) + \mathbb{E}_n(A|S = s), \forall s \in \mathcal{S}$.*

It is easy to see that $\mathcal{P}_O^{\mathcal{D}}(s^*, a^*) = a^* - \mathbb{E}_n(A|S = s^*) + \mathbb{E}_n(A)$ is a one-to-one function of $a^*$ for any fixed $s^*$. Denote $\hat{F}_{js}(x) = \mathbb{P}_n(A_j \leq x|S = s)$ as the empirical marginal cumulative distribution function (CDF) of the $j$th element of the non-sensitive attributes given the sensitive attribute $S = s$. Define its inverse as

$$\hat{F}_{js}^{-1}(z) = \inf\{x : \mathbb{P}_n(A_j \leq x|S = s) \geq z\}. \tag{3.1}$$

**Definition 3** (Marginal Distribution Mapping). *A marginal distribution mapping $\mathcal{P}_M^{\mathcal{D}}$ is a preprocessing procedure such that*

$$\mathcal{P}_M^{\mathcal{D}}(s^*, a^*) = \sum_s \hat{a}(s)\mathbb{P}_n(S = s),$$

*where the $j$th element of $\hat{a}(s)$ is $[\hat{a}(s)]_j = \hat{F}_{js}^{-1}(\hat{F}_{js^*}([a^*]_j))$ for $j = 1, \cdots, d$.*

Let $\mathcal{P}, \mathcal{P}_O$, and $\mathcal{P}_M$ denote the population level preprocessing procedure corresponding to $\mathcal{P}^{\mathcal{D}}, \mathcal{P}_O^{\mathcal{D}}$, and $\mathcal{P}_M^{\mathcal{D}}$, respectively. It is obvious that $\mathcal{P}_O(s^*, a^*) = a^* - \mathbb{E}(A|S = s^*) + \mathbb{E}(A)$ is still a one-to-one function of $a^*$ for any fixed $s^*$, and the $j$th element of $\mathcal{P}_M(s^*, a^*)$ is

$$[\mathcal{P}_M(s^*, a^*)]_j = \sum_s F_{js}^{-1}(F_{js^*}([a^*]_j))\mathbb{P}(S = s),$$

where $F_{js}$ is the marginal CDF of the $j$th element of $A$ given $S = s$ and $F_{js}^{-1}$ is defined similarly to (3.1) but replacing $\mathbb{P}_n$ with $\mathbb{P}$. It can be seen that if $A_j$ is a discrete variable, then $F_{js}^{-1}(F_{js^*}(x))$ is strictly increasing for $s = s^*$; and if $A_j$ is a continuous variable, then $F_{js}^{-1}(F_{js^*}(x))$ may not be strictly increasing when $F_{js^*}(x)$ is constant on some interval of $x$. Therefore $\mathcal{P}_M(s^*, a^*)$ is only a one-to-one function of $a^*$ for any fixed $s^*$ when the marginal CDF of each continuous element in $A$ given $S = s^*$ is strictly increasing.

### 3.1 Fair Learning Algorithm

Besides preprocessing procedures, we also have different choices of learners. A Fairness-Through-Unawareness (FTU) predictor $f_{FTU}(a)$ only uses the non-sensitive attributes $A$ to predict the conditional mean of $Y$. A Machine Learning predictor $f_{ML}(s,a)$ uses both the sensitive and non-sensitive attributes to predict $\mathbb{E}(Y|S, A)$. An Averaged Machine Learning (AML) predictor $f_{AML}(a) = \sum_s f_{ML}(s, a)\mathbb{P}_n(S = s)ds$. Note that we still need to train the ML predictor to obtain the AML predictor, but it only needs the non-sensitive attributes as its input when making a prediction since the sensitive attributes are averaged out. Algorithm 1 could use any learner $f \in \{f : \mathcal{A} \to [0,1]\}$ to learn the decisions from the processed data, and we would consider the FTU and AML learners in our numerical studies.

---

**Algorithm 1:** Fair Learning through dAta Preprocessing (FLAP)

**Input:** Training data $\mathcal{D}$, preprocessing procedure $\mathcal{P}^{\mathcal{D}}$, learner $f$, test attributes $(s, a)$.

1 **for** $(s_i, a_i, y_i)$ *in* $\mathcal{D}$ **do**
2     $a_i' = \mathcal{P}^{\mathcal{D}}(s_i, a_i)$
3 **end**
4 Create the processed data $\mathcal{D}' = \{(s_i, a_i', y_i)\}_{i=1}^n$
5 Learn predictor $f$ from $\mathcal{D}'$
6 Calculate $a' = \mathcal{P}^{\mathcal{D}}(s, a)$
7 Draw $\hat{Y}$ from Bernoulli($f(a')$)

**Output:** $\hat{Y}$

---

Apart from the structural assumptions made in Figure 1, extra conditions of the structural equation $f_A(s, u_A)$ must be satisfied for the preprocessing method to work.

**Condition 1** (Strong non-sensitive). *The partial derivative $\frac{\partial}{\partial u_A} f_A(s, u_A)$ does not involve $s$.*

**Condition 2** (Weak non-sensitive). *The sign of $\frac{\partial}{\partial u_A} f_{A_j}(s, u_A)$ does not change with $s$ for all $u_A$ and all $j = 1, \cdots, d$.*

These two conditions describe the relationship between the sensitive and non-sensitive attributes. Condition 2 is weaker than Condition 1. For example, an additive model $f_A(s, u_A) = \beta_0 + \beta_1 s + \beta_2 u_A$ satisfies both conditions, while an interaction model $f_A(s, u_A) = \beta_0 + \beta_1 s + \beta_2 u_A + \beta_3 s u_A$ does not satisfy Condition 1 but will satisfy Condition 2 if $\beta_2 + \beta_3 s$ is greater than (or less than, or equal to) zero for all $s$. In our running example, $\frac{\partial}{\partial u_A} f_A(s, u_A) = c_1 c_3 \sigma_a^s \exp\{c_2 + \lambda_a s + c_3 \sigma_a^s u_A\} > 0$ for $s = 0, 1$. So it meets Condition 2 but not Condition 1. We prove in the following theorem that these conditions, together with the SCM, are sufficient for Algorithm 1 to generate counterfactually fair decisions.

**Theorem 1.** *Let $\hat{Y}$ be the output from Algorithm 1, i.e., $\mathbf{1}\{U_{\hat{Y}} < f(\mathcal{P}^{\mathcal{D}}(s, a))\}$.*

1. *If the procedure $\mathcal{P}_O^{\mathcal{D}}$ is adopted, $\hat{Y}$ is counterfactually fair under Condition 1.*

2. *If the procedure $\mathcal{P}_M^{\mathcal{D}}$ is adopted, $\hat{Y}$ is counterfactually fair under Condition 2.*

We prove Theorem 1 in Appendix A. The intuition is that the FLAP algorithm learns the decision from processed data only, and the processed data contain no sensitive information since the preprocessing procedure can remove $A$'s dependence on $S$ under the non-sensitive condition.

### 3.2 Test for Counterfactual Fairness

Data preprocessing not only allows us to learn a counterfactually fair decision but also enables us to test if the decisions made in the original data are fair. When Condition 1 holds, we can use the data processed by the orthogonalization procedure to test fairness. When the strong condition does not hold but Condition 2 is satisfied, we need an extra condition to utilize the marginal distribution mapping procedure for fairness testing.

**Condition 3.** *The conditional marginal CDF $F_{js}(x)$ is strictly increasing for all such $j$ that $A_j$ is continuous and all $s \in \mathcal{S}$.*

In other words, each non-sensitive attributes $A_j$ should be either a discrete random variable or a continuous one with non-zero density on $\mathbb{R}$. This condition ensures that $\mathcal{P}_M(s^*, a^*)$ is a one-to-one function as discussed earlier. With these conditions, we can establish the equivalence between CF and the conditional independence of decision and sensitive information given the processed non-sensitive information.

**Theorem 2.** *Consider the original decision $Y$:*

1. *Under Condition 1, $Y$ is counterfactually fair if and only if $Y \perp S | \mathcal{P}_O(S, A)$.*

2. *Under Conditions 2 and 3, $Y$ is counterfactually fair if and only if $Y \perp S | \mathcal{P}_M(S, A)$.*

Its proof is in Appendix A. Theorem 2 allows us to test CF using any well-established conditional independence test. In practice, given a decision dataset $\mathcal{D} = (s_i, a_i, y_i)_{i=1}^n$, we can obtain the empirical processed non-sensitive attributes $\mathcal{P}^{\mathcal{D}}(s_i, a_i)$ and test if $Y \perp S | \mathcal{P}^{\mathcal{D}}(S, A)$. If the p-value of the test is small enough for us to reject the conditional independence hypothesis, then the original decision is probably biased and algorithms such as FLAP should be used to learn fair decisions.

## 4 NUMERICAL STUDIES

In this section, we compare the decisions made by different algorithms in terms of fairness and accuracy using simulated and real data, and also investigate the empirical performance of the fairness test using simulated data with small sample sizes. We consider three cases for generating the simulation data. The first one is Example 1 and the second one is a multivariate extension of it where we introduce one more sensitive group and include the education years of the loan applicants as another non-sensitive attribute and let their annual income depend on it. The third example is a replica of the admission example constructed by Wang et al. (2019). The details of these examples and the parameters chosen in the simulation are presented in Appendix B.

As discussed before, Condition 2 is satisfied in Example 1 but Condition 1 is not. Moreover, both Examples 2 and 3 do not satisfy either condition in general due to the cutoff in the value of their non-sensitive attributes, and hence neither of the proposed preprocessing methods can achieve CF in theory. However, the weaker Condition 2 will hold in Example 2 when the mean education years of the three sensitive groups are the same, in which case the marginal distribution mapping method should work.

### 4.1 FAIRNESS EVALUATION

We compare our FLAP algorithm with

1. ML: the machine learning method using both sensitive and non-sensitive attributes without preprocessing, which is a logistic regression of $Y$ on $S$ and $A$;

2. FTU: the Fairness-Through-Unawareness method which fits a logistic model of $Y$ on non-sensitive attributes $A$ alone without preprocessing;

3. FL: the FairLearning algorithm proposed by Kusner et al. (2017);

4. AA: the Affirmative Action algorithm proposed by Wang et al. (2019).

All these methods can output a predicted score $p$ given the training data $\mathcal{D}$ and test attributes $(s, a)$, denoted $p(s, a; \mathcal{D})$ and draw the random decision $\hat{Y}$ from $\text{Bernoulli}(p(s, a; \mathcal{D}))$. For ML method, $p(s, a; \mathcal{D}) = f_{ML}(s, a)$; for FTU method, that is $f_{FTU}(a)$. We denote the predicted scores of the FairLearning and AA algorithms as $f_{FL}(s, a; \mathcal{D})$ and $f_{AA}(s, a; \mathcal{D})$, respectively. For our FLAP method, we use the marginal distribution mapping procedure and try both the AML and the FTU learners described in Section 3 and name the methods as FLAP-1 and FLAP-2. Their predicted scores are $f_{AML}(\mathcal{P}_M^{\mathcal{D}}(s, a))$ and $f_{FTU}(\mathcal{P}_M^{\mathcal{D}}(s, a))$, respectively. We mainly use the Mean Absolute Error (MAE) of the predicted score on the test set to measure the prediction performance, and also include the area under the ROC curve and average precision of the prediction in Appendix B for

completeness. All these metrics show similar results about the prediction performance. The metric for measuring the counterfactual fairness (CF-metric) is defined as

$$\max_{r,t\in\mathcal{S}} \frac{1}{N_{test}} \sum_{i=1}^{N_{test}} |p(r,\hat{a}_M^{\mathcal{D}}(r,s_i,a_i);\mathcal{D}) - p(t,\hat{a}_M^{\mathcal{D}}(t,s_i,a_i);\mathcal{D})|,$$

where $\hat{a}_M^{\mathcal{D}}(s,s^*,a^*)$ is defined as $\hat{a}(s)$ in Definition 3. Note that the CF-metric should be zero when decisions are CF under Condition 2. In real world applications where the condition cannot be verified, a CF decision is not guaranteed to have zero CF-metric, although we expect the CF-metric to be lower for fairer decisions in general. This definition is different from the AA-metric proposed by Wang et al. (2019) in two folds. First, it allows us to consider more than two sensitive groups by taking the maximum of the pairwise difference of predicted scores, but it reduces to the AA-metric for two sensitive groups. Second, we use the marginal distribution mapping method to compute the counterfactual non-sensitive attributes $\hat{a}_M^{\mathcal{D}}(s,s^*,a^*)$ had the unit been in a different sensitive group $s$. This ensures that all the derived counterfactual attributes are within the range of observed attribute values. In comparison, Wang et al. (2019) use the orthogonalization method to compute the counterfactual attributes and thus a female student having test score $0.98$ would have a counterfactual score of $1.48$ had she been a male if the male mean test score is $0.5$ higher than female. This out-of-range counterfactual score is unreasonable and problematic when being used as the input of the score prediction function $p$.

For Example 1, we hold other parameters fixed while increase $\sigma_a$ from $1$ to $2.8$ to see how the difference in the variation of the non-sensitive attribute between sensitive groups affects fairness. As expected, the AA algorithm which essentially uses the orthogonalization method cannot achieve CF since Condition 1 is not met. However, both FLAP algorithms' CF-metrics are zero when using the marginal distribution mapping preprocessing (Figure 2a).

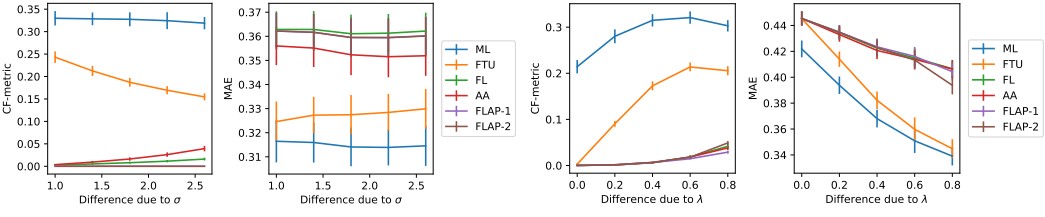

(a) Example 1 with increasing $\sigma_a$.     (b) Example 3 with different mean scores by gender.

Figure 2: Comparison of CF-metric and MAE of decision making algorithms

Wang et al. (2019) showed that the AA algorithm can achieve zero AA-metric in Example 3, but it does not satisfy either of the non-sensitive conditions for achieving CF. As shown in Figure 2b, all algorithms we consider cannot achieve CF, but the FLAP algorithms still have the lowest CF-metric. The results of Example 2 are shown in Appendix B and there is no significant difference between the MAE of the AA and FLAP algorithms in all examples. In general, we expect fairer predictions to have higher MAEs since they correct the discriminatory bias of the original decisions.

## 4.2 FAIRNESS TEST

The Conditional Distance Correlation (CDC) test (Wang et al., 2015) is a well-established non-parametric test for conditional independence. We use it here to illustrate the performance of the fairness test with the three simulated examples. For each example, we use different combinations of parameters to obtain simulated datasets with different fairness levels, which are measured by the CF-metric. A CDC test with a significance level of $0.05$ is then conducted to test if $Y \perp S | \mathcal{P}^{\mathcal{D}}(S,A)$ for each dataset. The simulation-test process is repeated 1000 times for each combination of parameters to estimate the power of the test, namely the probability of rejecting the null hypothesis that the decisions are counterfactually fair. The results are summarized in Figure 3.

When the decisions are generated fair, which are shown as the points with CF-metrics equal to zero, the type I error rate is around $0.05$ for all examples. The power of the test grows as we make the decisions more unfair, or increase the sample size.

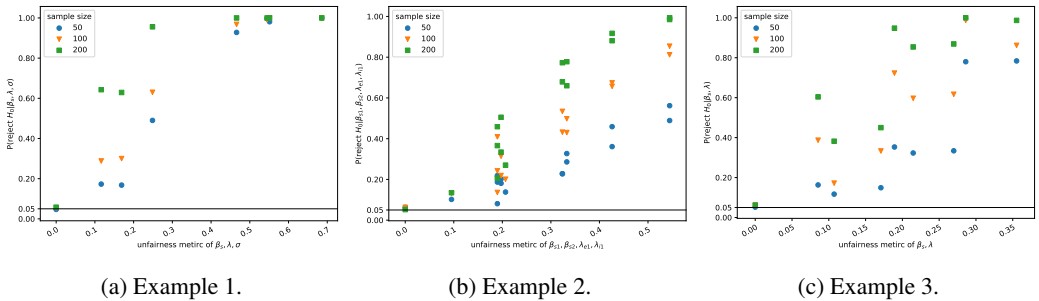

| (a) Example 1. | (b) Example 2. | (c) Example 3. |

Figure 3: Power for testing CF using conditional independent test plot against the CF-metric

## 5 REAL DATA ANALYSIS

We apply our methods to a loan application dataset from a fintech company, the adult income dataset from UCI Machine Learning Repository[1] and the COMPAS recidivism data from ProPublica[2] (Angwin & Larson, 2016). Here we present our analysis of the loan application data, and the results from the other two publicly available datasets are shown in Appendix C since the conclusions are similar.

In the loan application case, the fintech lender aims to provide short-term credit to young salaried professionals by using their mobile and social footprints to determine their creditworthiness even when a credit history may not be available. To get a loan, a customer has to download the lending app, submit all the requisite details and documentation, and give permission to the lender to gather additional information from her/his smartphone, such as the number of apps, number of calls, and SMSs, and number of contacts and social connections. We obtained data from the lending firm for all loans granted from February 2016 to November 2018. The decisions $Y$ are whether or not the lender approves the loan applications. The attributes are applicants' gender, age, salary, and other information collected from their smartphones. Both gender and age are regarded as sensitive information here and we find that the decisions are made in favor of the senior and female applicants. Since we can only deal with categorical sensitive attributes, we divide the applicants into two age groups by the lower quartile of the age distribution and create a categorical variable $S \in \{0, 1, 2, 3\}$ to denote the group of the applicants: female younger than 28; male younger than 28; female older than 28; and male older than 28. The effective sample size after removing missing values is 203,656.

Non-parametric conditional independence tests will not be efficient for this real case due to the large sample size. Therefore we test the conditional independence of $Y$ and $S$ given $\mathcal{P}_M^{\mathcal{D}}(S, A)$ by fitting a simple logistic model for $Y$ with $S$ and $\mathcal{P}_M^{\mathcal{D}}(S, A)$ as the explanatory variables and testing if the coefficient of $S$ is significantly different from zero. The p-value of the F-test is almost zero and indicates that the decisions are unfair for applicants in different groups. When other attributes are fixed to their means, the predicted approval probabilities of the four groups from the logistic model are 0.924 (young female), 0.899 (young male), 0.948 (senior female), and 0.946 (senior male), also indicating that the decisions are most in favor of the senior and female applicants.

We then separate the data into a training set of 193,656 samples and a test set of 10,000 samples. The training dataset is used to learn the decisions with different algorithms and the test dataset is used to evaluate the CF-metric and MAE. The results are summarized in Table 1. Our FLAP algorithms have lower CF-metrics compared with other algorithms and their MAEs are only greater than the ML method. Among the FLAP algorithms, the two using the marginal distribution mapping preprocessing procedure have better CF-metric and similar MAE. The FLAP algorithm using the FTU learner (FLAP-2) performs slightly better than the one using the AML learner (FLAP-1). Note that in real-world applications, fairer decisions may not have higher MAEs as expected in the simulation studies because we do not have access to all the variables possessed by the original decision-maker. When the original decisions depend on additional information, the FLAP and other

---

[1] https://archive.ics.uci.edu/ml/machine-learning-databases/adult/
[2] https://github.com/propublica/compas-analysis

fair learning methods may yield predictions closer to or further away from the original decisions, and thus leading to lower or higher MAEs.

Table 1: Comparison of the CF-metric and MAE of decision making algorithms on the fintech data. FLAP-1(O) and FLAP-2(O) use the orthogonalization and FLAP-1(M) and FLAP-2(M) use the marginal distribution mapping preprocessing procedure.

|  | ML | FTU | FL | AA | FLAP-1(O) | FLAP-2(O) | FLAP-1(M) | FLAP-2(M) |
|---|---|---|---|---|---|---|---|---|
| CF-metric | 0.0392 | 0.0130 | 0.0011 | 0.0011 | 0.0011 | 0.0011 | 0.0008 | 0.0007 |
| MAE | 0.1249 | 0.1261 | 0.1258 | 0.1266 | 0.1258 | 0.1258 | 0.1258 | 0.1258 |

## 6 DISCUSSION

We propose two data preprocessing procedures and the FLAP algorithm to make counterfactually fair decisions. The algorithm is general enough so that any learning methods from logistic regression to neural networks can be used, and counterfactual fairness is guaranteed regardless of the learning methods. The orthogonalization procedure is faster and ensures counterfactually fair decisions when the strong non-sensitive condition is met. The marginal distribution mapping procedure is more complex but guarantees fairness under the weaker non-sensitive condition.

We also prove the equivalence between counterfactual fairness and the conditional independence of decisions and sensitive attributes given the processed non-sensitive attributes under the non-sensitive assumptions. We illustrate that the CDC test is reliable for testing counterfactual fairness when the sample size is small. When the size gets bigger, however, we need a more efficient testing method for the fairness test.

It is well understood but still worth noting that causal inference comes with strong assumptions, such as the SCM and the non-sensitive conditions in our case. Moreover, these assumptions are often unverifiable in general, although we may test some of them when only considering a restricted class of models. As the saying goes, "all models are wrong, but some are useful". The FLAP method may require unverifiable assumptions in practice, but we make it general enough and easy to follow in the hope that this would encourage decision-makers to address the fairness issue with its help.

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

# A  PROOF OF MAIN RESULTS

## A.1  PROOF OF THEOREM 1

*Proof.* We prove the theorem for a general class of learners $\{f : \mathcal{A} \to [0, 1]\}$ that only take the non-sensitive attribute $a$ as the input. Clearly, both $f_{FTU}$ or $f_{AML}$ belong to this class. We follow the Abduction-Action-Prediction steps in Theorem 7.1.7 Pearl (2009b) to evaluate the conditional expectation of $\hat{Y}_{s'}(U)$ given the evidence $S = s^*, A = a^*$,

$$\mathbb{E}(\hat{Y}_{s'}(U)|S = s^*, A = a^*) = \int f(\mathcal{P}^{\mathcal{D}}(s', f_A(s', u))) p_{U_A|S,A}(u|S = s^*, A = a^*) du,$$

where $p_{U_A|S,A}(u|s^*, a^*)$ denotes the conditional density of $U_A$ given $S = s^*$ and $A = a^*$. If $\mathcal{P}^{\mathcal{D}}(s', f_A(s', u))$ does not depend on $s'$, so will $\mathbb{E}(\hat{Y}_{s'}(U)|S = s^*, A = a^*)$ and we will have

$$\mathbb{E}(\hat{Y}_{s'}(U)|S = s^*, A = a^*) = \mathbb{E}(\hat{Y}_{s^*}(U)|S = s^*, A = a^*).$$

Note that $\mathcal{P}^{\mathcal{D}}(s', f_A(s', u)) = \sum_s \hat{a}(s) \mathbb{P}_n(S = s)$ for both the preprocessing procedures we are considering. Therefore, it suffices to show that $\hat{a}(s)$ does not depend on $s'$.

First, consider the Orthogonalization procedure $\mathcal{P}_O^{\mathcal{D}}$ where

$$\hat{a}(s) = f_A(s', u) - \mathbb{E}_n(A|S = s') + \mathbb{E}_n(A|S = s)$$
$$= f_A(s', u) - \mathbb{E}(A|S = s') + \mathbb{E}_n(A|S = s) - (\mathbb{E}_n - \mathbb{E})(A|S = s').$$

Note that $A|\{S = s'\} = f_A(s', U_A)$ and the first order Taylor expansion of it gives

$$\mathbb{E}(A|S = s') = \mathbb{E}\left( f_A(s', u) + \frac{\partial}{\partial u} f_A(s, u)\Big|_{s=s', u=u'} (U_A - u) \right)$$
$$= f_A(s', u) + \frac{\partial}{\partial u} f_A(s, u)\Big|_{s=s', u=u'} \mathbb{E}(U_A - u)$$

for some $u'$ between $u$ and $U_A$. By Condition 1

$$\hat{a}(s) = \frac{\partial}{\partial u} f_A(s, u)\Big|_{s=s^*, u=u'} \mathbb{E}(u - U_A) + \mathbb{E}_n(A|S = s) + o_{\mathbb{P}}(n)$$

and thus it does not depend on $s'$.

Second, consider the Marginal Distribution Mapping procedure $\mathcal{P}_M^{\mathcal{D}}$. Let $f_{A_j}(s, u) = e_j^T f_A(s, u)$ where $e_j$ is a $d$-dimensional vector with the $j$th element being one and all other elements being zeros. The $j$th element of $\hat{a}(s)$ is $[\hat{a}(s)]_j = \hat{F}_{js}^{-1}(\hat{F}_{js'}(f_{A_j}(s', u)))$ for $j = 1, \cdots, d$. Again, the first order Taylor expansion of $f_{A_j}(s', U_A)$ gives

$$\hat{F}_{js'}(f_{A_j}(s', u)) = \mathbb{P}_n(A_j \le f_{A_j}(s', u)|S = s')$$
$$= \mathbb{P}(f_{A_j}(s', U_A) \le f_{A_j}(s', u)) + (\mathbb{P}_n - \mathbb{P})(A_j \le f_{A_j}(s', u)|S = s')$$
$$= \mathbb{P}\left( f_{A_j}(s', u) + \frac{\partial}{\partial u} f_{A_j}(s, u)\Big|_{s=s', u=u'} (U_A - u) < f_{A_j}(s', u) \right) + o_{\mathbb{P}}(n)$$

for some $u'$ between $u$ and $U_A$. Under Condition 2,

$$\hat{F}_{js'}(f_{A_j}(s', u)) = \mathbb{P}\left( \text{sign}\left( \frac{\partial}{\partial u} f_{A_j}(s, u)\Big|_{s=s^*, u=u'} \right) (U_A - u) < 0 \right) + o_{\mathbb{P}}(n)$$

does not depend on $s'$ and hence $a(s)$ is a function of $s$ and $u$ alone. □

## A.2  PROOF OF THEOREM 2

*Proof.* The steps of proving the two statements of Theorem 2 are similar. To remove redundancy, we use the notation $\mathcal{P}$ whenever the argument is true for both the preprocessing procedures $\mathcal{P}_O$ and $\mathcal{P}_M$.

First, we show that $Y$ is counterfactually fair if $Y \perp S | \mathcal{P}(S, A)$. The posterior mean of the counterfactual $Y_{s'}(U)$ given $S = s^*$ and $A = a^*$ can be evaluated in two steps: first find the conditional distribution of $U = \{U_A, U_S, U_Y\}$, and then calculate the conditional expectation of the counterfactuals from the SCM. Since the effect of $U_S$ is blocked by setting $S = s'$ and $U_Y$ is independent of $S$ and $A$, only the distribution of $U_A$ will be affected by the given information and effect the counterfactuals $Y_{s'}(U)$.

$$\mathbb{E}(Y_{s'}(U)|S = s^*, A = a^*) = \int \mathbb{E}(f_Y(s', f_A(s', u), U_Y) p_{U_A|S,A}(u|S = s^*, A = a^*)du. \quad \text{(A.1)}$$

Under the SCM, $\mathbb{E}(f_Y(s', f_A(s', u), U_Y)$ is the same as the expectation of the observed decision $Y$ given the attributes $S = s', A = f_A(s', u)$. Therefore (A.1) is equal to

$$\int \mathbb{E}(Y|S = s', A = f_A(s', u)) p_{U_A|S,A}(u|S = s^*, A = a^*)du \quad \text{(A.2)}$$

$$= \int \mathbb{E}(Y|S = s', \mathcal{P}(S, A) = \mathcal{P}(s', f_A(s', u))) p_{U_A|S,A}(u|S = s^*, A = a^*)du \quad \text{(A.3)}$$

$$= \int \mathbb{E}(Y|\mathcal{P}(S, A) = \mathcal{P}(s', f_A(s', u))) p_{U_A|S,A}(u|S = s^*, A = a^*)du \quad \text{(A.4)}$$

$$= \int \mathbb{E}(Y|\mathcal{P}(S, A) = \mathcal{P}(s^*, f_A(s^*, u))) p_{U_A|S,A}(u|S = s^*, A = a^*)du \quad \text{(A.5)}$$

$$= \mathbb{E}(Y_{s^*}(U)|S = s^*, A = a^*). \quad \text{(A.6)}$$

Equation (A.3) replaces the condition $A = f_A(s', u)$ with $\mathcal{P}(S, A) = \mathcal{P}(s', f_A(s', u))$ because $\mathcal{P}_O(S, A)$ is a one-to-one function of $A$ given $S$ and $\mathcal{P}_M(S, A)$ is also a one-to-one function of $A$ given $S$ under Condition 3. Equation (A.4) is due to the conditional independence of $Y$ and $S$, and (A.5) uses the result that $\mathcal{P}(s', f_A(s', u)) = \mathcal{P}(s^*, f_A(s^*, u))$, which can be shown following the proof of Theorem 1. Repeat the steps (A.1) to (A.5) and we shall get the same result for $\mathbb{E}(Y_{s'}(U)|S = s^*, A = a^*)$. Note that both $Y_{s'}(U)$ and $Y_{s^*}(U)$ are binary random variables, therefore the equivalence in expectation implies that

$$Y_{s'}(U)|\{S = s^*, A = a^*\} \overset{d}{=} Y_{s^*}(U)|\{S = s^*, A = a^*\}.$$

The above result holds for any $s', s^* \in \mathcal{S}$, so the definition of counterfactual fairness is satisfied.

Next we show that $Y \perp S | \mathcal{P}(S, A)$ if $Y$ is counterfactually fair. The counterfactual fairness of $Y$ implies

$$\begin{aligned} &\mathbb{E}[f_Y(s', f_A(s', U_A), U_Y)|S = s^*, A = a^*] \\ =&\mathbb{E}[f_Y(s^*, f_A(s^*, U_A), U_Y)|S = s^*, A = a^*]. \end{aligned} \quad \text{(A.7)}$$

Let $a' = \mathcal{P}(s^*, a^*)$, then

$$(U_A, U_Y)|\{S = s^*, A = a^*\} \overset{d}{=} (U_A, U_Y)|\{S = s^*, \mathcal{P}(s^*, A) = a'\} \quad \text{(A.8)}$$

since $\mathcal{P}(s^*, a^*)$ is a one-to-one function of $a^*$ for each $s^*$.

Using the Bayesian formula, the posterior density of $U_A$ is

$$p_{U_A|S,\mathcal{P}}(u|S = s^*, \mathcal{P}(s^*, A) = a') \quad \text{(A.9)}$$

$$= \frac{p_{U_A}(u)\mathbb{P}(S = s^*)p_{\mathcal{P}|S,U_A}(a'|S = s^*, U_A = u)}{\mathbb{P}(S = s^*)\int p_{U_A}(u)p_{\mathcal{P}|S,U_A}(a'|S = s^*, U_A = u)du}, \quad \text{(A.10)}$$

where $p_{U_A|S,\mathcal{P}}(u|s^*, a')$ denotes the conditional density of $U_A$ given $S = s^*$ and $\mathcal{P}(s^*, A) = a'$, $p_{U_A}(u)$ denotes the prior density of $U_A$, and $p_{\mathcal{P}|S,U_A}(a'|S = s^*, U_A = u)$ denotes the conditional density of $\mathcal{P}(s^*, A)$ given $S = s^*$ and $U_A = u$. As a density function, (A.10) is proportional to its kernel $p_{U_A}(u)p_{\mathcal{P}|S,U_A}(a'|S = s^*, U_A = u)$, which equals $p_{U_A}(u)p_{\mathcal{P}|S,U_A}(a'|S = s', U_A = u)$ because $\mathcal{P}(S, A)$ does not depend on $S$ when $U_A$ is given as shown in the proof of Theorem 1. Repeating the steps and we can show that the posterior density of $U_A$ given $S = s', \mathcal{P}(s', A) = a'$ is also proportional to $p_{U_A}(u)p_{\mathcal{P}|S,U_A}(a'|S = s', U_A = u)$. Together with the assumption in the SCM that $U_Y$ is independent of $S, \mathcal{P}(S, A)$, we have

$$(U_A, U_Y)|\{S = s', \mathcal{P}(s', A) = a'\} \overset{d}{=} (U_A, U_Y)|\{S = s^*, \mathcal{P}(s^*, A) = a'\}. \quad \text{(A.11)}$$

The intuition here is that if the processed non-sensitive data are equal, then they provide the same information about $U_A$ regardless of the sensitive information in the original data. Substituting the conditions $\{S = s^*, A = a^*\}$ in (A.7) with the equivalent conditions in (A.8) and (A.11) gives

$$
\begin{aligned}
&\mathbb{E}[f_Y(s', f_A(s', U_A), U_Y)|S = s', \mathcal{P}(s', A) = a'] \\
=&\mathbb{E}[f_Y(s^*, f_A(s^*, U_A), U_Y)|S = s^*, \mathcal{P}(s^*, A) = a'].
\end{aligned}
\tag{A.12}
$$

Under the SCM and structural equations defined in Figure 1, (A.12) implies

$$
\mathbb{E}[Y|S = s', \mathcal{P}(S, A) = a'] = \mathbb{E}[Y|S = s^*, \mathcal{P}(S, A) = a'].
\tag{A.13}
$$

Since (A.13) holds for any $s', s^* \in \mathcal{S}$, it yields that

$$
\mathbb{E}[Y|S, \mathcal{P}(S, A)] = \mathbb{E}[Y|\mathcal{P}(S, A)]
$$

and hence $Y \perp S|\mathcal{P}(S, A)$ for binary $Y$. □

## B  DETAILS OF SIMULATION EXAMPLES

**Example 2.** *The bank now collects the race $S$, education year $E$ and annual income $A$ information from loan applicants. There are three possible race groups $\mathcal{S} = \{0, 1, 2\}$ and $S = \mathbf{1}\{U_S > 0.76\} + \mathbf{1}\{U_S > 0.92\}$, meaning that a random applicant could be from the majority race group (0) with probability 0.76, or from the minority group 1 or 2 with probability 0.16 or 0.08. Let $U_E$ be a standard normal random variable and $\mu_E = \lambda_{e0} + \mathbf{1}\{S = 1\}\lambda_{e1} + \mathbf{1}\{S = 2\}\lambda_{e2}$, the education year is*

$$
E = \max\{0, \mu_E + 0.4\mu_E U_E\}.
$$

*Let $\mu_A = \log(\lambda_{a0} + \mathbf{1}\{S = 1\}\lambda_{a1} + \mathbf{1}\{S = 2\}\lambda_{a2})$, the annual income is*

$$
A = \exp\{\mu_A + 0.4\mu_E U_E + 0.1 U_A\}.
$$

*The decision of the bank is modeled as*

$$
Y = \mathbf{1}\{U_Y < \mathrm{expit}(\beta_0 + \mathbf{1}\{S = 1\}\beta_1 + \mathbf{1}\{S = 2\}\beta_2 + \beta_a A + \beta_e E)\}.
$$

Here $\lambda_{e0}, \lambda_{e1}$, and $\lambda_{e2}$ decide the mean education year of the three race groups. $\lambda_{a0}, \lambda_{a1}$, and $\lambda_{a2}$ decide the median annual income. The annual income and the education year are positively correlated through $U_E$. $\beta_1$ and $\beta_2$ characterize the direct effect of the race information while the $\lambda$'s indicate the indirect effect together with $\beta_e$ and $\beta_a$. In this example, neither of Conditions 1 and 2 holds if $\beta_e$ and $\lambda_{e1}$ and/or $\lambda_{e2}$ are not zero due to the maximum operator in $f_E$. Even if $\lambda_{e1} = \lambda_{e2} = 0$, only the weaker Condition 2 will hold due to the same reason for Example 1.

Example 3 is a replica of the admission example constructed by Wang et al. (2019).

**Example 3.** *The admission committee of a university collects the gender $S$ and test score $T$ information from applicants. The gender is simulated from $S = \mathbf{1}\{U_S < 0.5\}$, where $S = 1$ for male and $S = 0$ for female. Let $U_T \sim \mathrm{Uniform}(0, 1)$ and we generate the test score as*

$$
T = \min\{\max\{0, \lambda S + U_T\}, 1\}.
$$

*The decision of the committee is*

$$
Y = \mathbf{1}\{U_Y < \mathrm{expit}(\beta_0 + \beta_t T + \beta_s S)\}.
$$

It is worth noting that both Examples 2 and 3 do not satisfy either of Conditions 1 and 2 due to the cutoff in the value of their non-sensitive attributes education years and test score. Take Example 3, there will be a positive probability ($\lambda$ to be exact) of seeing male students with test score 1 if $\lambda > 0$. Check that

$$
\frac{\partial}{\partial u_T} f_T(s, u_T) = \begin{cases} 1, & 0 < u_T < 1 - \lambda s \\ 0, & 1 - \lambda s < u_T < 1 \end{cases}
$$

and we can see that its sign does change with $s$ for any fixed $u_T$. Therefore, neither of the proposed preprocessing methods can achieve CF in theory.

In Figures 2a and 4c, we choose $c_1 = 0.01$, $c_2 = 4$, $c_3 = 0.2$ and fix $\beta_0 = -1$, $\beta_a = 2$, $\beta_s = 1$, $\lambda_a = 0.5$ while increase $\sigma_a$ from 1 to 2.8 to see how the difference in the variation of non-sensitive

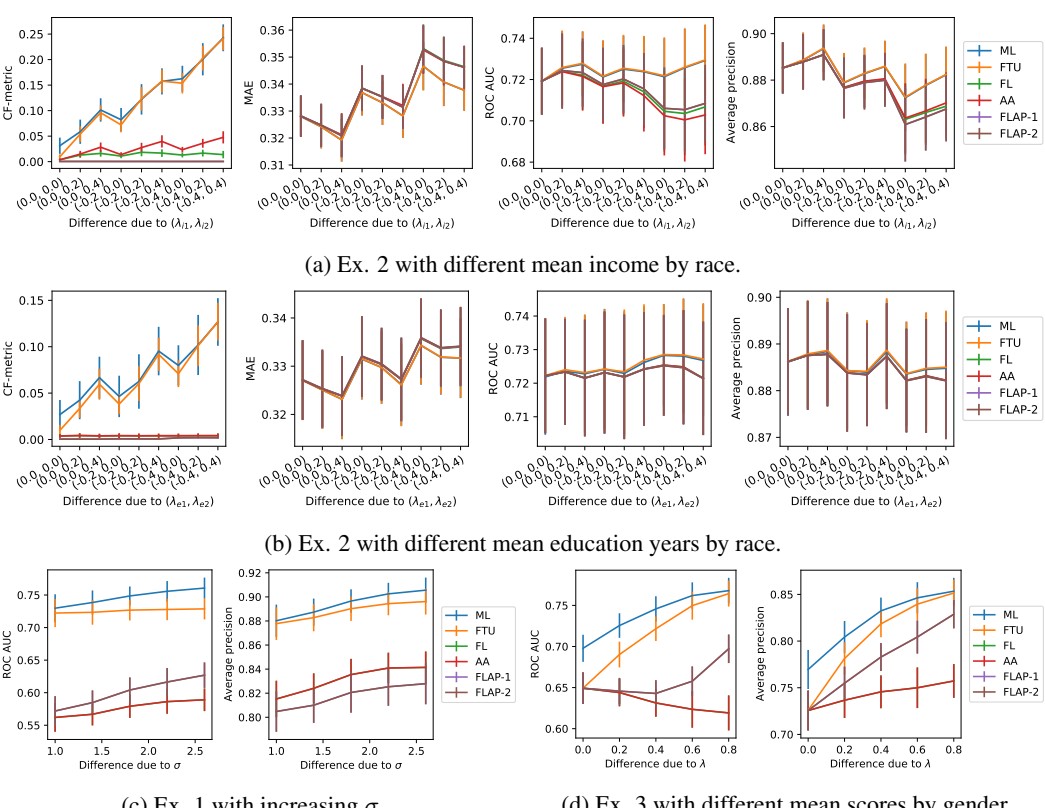

(a) Ex. 2 with different mean income by race.

(b) Ex. 2 with different mean education years by race.

(c) Ex. 1 with increasing $\sigma_a$.

(d) Ex. 3 with different mean scores by gender.

Figure 4: Comparison of CF-metric and prediction performance of decision making algorithms

attribute between sensitive groups affects fairness. In Figures 2b and 4d, we set $\beta_0 = -1$, $\beta_t = 2$, $\beta_s = 1$ and increase $\lambda$ from 0 to 0.8 to see how the mean difference of test scores affects fairness.

In Figure 4a and 4b, we choose $\beta_0 = -1$, $\beta_1 = \beta_2 = 0$, $\beta_a = 1$, $\beta_e = 2$, $\lambda_{e0} = 1.07$, $\lambda_{i0} = 0.58$. In Figure 4a, we change $(\lambda_{i1}, \lambda_{i2})$ while fix $\lambda_{e1} = 0$ and $\lambda_{e2} = 0$ to see how the mean difference of income affect fairness. The results are telling the same story as Figure 2a: since only the weaker non-sensitive condition is met, the AA-algorithm cannot achieve CF but the FLAP algorithms with marginal distribution mapping procedure can.

In Figure 4b, we change $(\lambda_{e1}, \lambda_{e2})$ while fix $\lambda_{i1} = 0$ and $\lambda_{i2} = 0$ to see how the mean difference of education affect fairness. The results are similar to those of Figure 2b where all algorithms we consider cannot achieve CF but the FLAP algorithms still have the lowest CF-metric.

## C    DETAILS OF REAL DATA ANALYSIS

We use the adult income data to predict whether an individual's income is higher than \$50K with information including sex, race, age, workclass, education, occupation, marital-status, capital gain and loss. Sex and race are regarded as sensitive attributes. The training set has 32,561 samples and the test set has 16281 samples. The comparison of the FLAP and other methods are shown in Table 2.

The COMPAS (Correctional Offender Management Profiling for Alternative Sanctions) recidivism data contains the demographic data such as sex, age, race, and record data such as priors count, juvenile felonies count, and juvenile misdemeanors count of over 10,000 criminal defendants in Broward County, Florida. The task is to predict whether they will re-offend in two years. According to ProPublica, "Black defendants were often predicted to be at a higher risk of recidivism than they actually were." Here we treat sex and race as sensitive attributes and try to predict recidivism in a counterfactually fair manner. We only use the data for Caucasian, Hispanic, and African-American individuals due to the small sample sizes of other races. The remaining data are divided into a training set of 5,090 samples and a test set of 1697 samples. The results are shown in Table 3

Table 2: Comparison of the CF-metric and MAE of decision making algorithms on the adult income data. FLAP-1(O) and FLAP-2(O) use the orthogonalization and FLAP-1(M) and FLAP-2(M) use the marginal distribution mapping preprocessing procedure.

|  | ML | FTU | FL | AA | FLAP-1(O) | FLAP-2(O) | FLAP-1(M) | FLAP-2(M) |
|---|---|---|---|---|---|---|---|---|
| CF-metric | 0.2779 | 0.2338 | 0.0228 | 0.0268 | 0.0280 | 0.0228 | 0.0020 | 0.0022 |
| MAE | 0.2388 | 0.2396 | 0.2406 | 0.2356 | 0.2452 | 0.2406 | 0.2430 | 0.2401 |

Table 3: Comparison of the CF-metric and MAE of decision making algorithms on the COMPAS data. FLAP-1(O) and FLAP-2(O) use the orthogonalization and FLAP-1(M) and FLAP-2(M) use the marginal distribution mapping preprocessing procedure.

|  | ML | FTU | FL | AA | FLAP-1(O) | FLAP-2(O) | FLAP-1(M) | FLAP-2(M) |
|---|---|---|---|---|---|---|---|---|
| CF-metric | 0.2274 | 0.1406 | 0.0054 | 0.0060 | 0.0058 | 0.0054 | 0.0026 | 0.0027 |
| MAE | 0.4256 | 0.4274 | 0.4402 | 0.4391 | 0.4395 | 0.4401 | 0.4393 | 0.4393 |

Similar to the results we have shown for the loan application data, the FLAP methods using the marginal distribution mapping preprocessing procedure have lower CF-metric and are thus considered fairer than other fair learning algorithms, and their MAEs are comparable to other methods.

