# OpenReview forum: "Counterfactual Fairness through Data Preprocessing"
_ICLR.cc/2021/Conference — Reject_

### Official Review · AnonReviewer1 · 2020-10-27
**An interesting paper but some poitn need to be clarified**

**Rating:** 5
**Confidence:** 3

**Review:**

The authors propose a preprocessing method that eases fair learning called FLAP.
They focus on counterfactual fairness introduced by Kusner et al. 2017. Under certain conditions, the preprocessing allow to biased the data such that traditional learning becomes fair.
This is a very interesting idea that is novel to me.

The paper is well written, and the related work is coherent with the work done. However, I think a few new paper are missing, for example Counterfactual Fairness: Unidentification, Bound and Algorithm from Wu et al 2019 that bounded the reachable counterfactual fairness. It would have been interesting to see how the results related to these bounds.

I think that the statement:  “We prove that CF is equivalent to the conditional independence of the decision and the sensitive attributes given the processed non-sensitive attributes” is a bit misleading. As far as I understand, this is only true given that either condition 1 is satisfied or conditions 2 and 3 are satisfied. There exists statistical test for conditional independence but checking for condition 1, 2, 3 seems more complicated.

On the real data, I would like to see a discussion on the meaning of CF-metric and FLAP method given that the condition 1, 2,3 are not verified.

I think that theses clarifications are needed for a clear accept.

----- edit after the authors' answers -----
I'm not convinced by the authors' answer on the meaning of the CF-metric when condition 2 is not verified, which is the case on real data. "... but we would expect fairer decisions to have smaller CF-metric in practice" doesn't seem enough to claim that FLAP is better because its CF-metric is lower. In addition we don't have confidence interval to assess if FLAP is really better on MAE.

Finally the review of reviewer1 made me realize that CF-metric and FLAP are based on the same function. I partly agree with the authors' answer, i.e., we need a counterfactual function to construct a good metric and then it is optimal to use it as well in the algorithm. However it is not fair to use that metric to claim that the algorithm is fairer.

Therefore I lower my rating to 5.

---

> ### Author Response · Authors · 2020-11-22
> **Response to Reviewer1**
>
> We sincerely thank you for the encouraging and constructive comments. In this revision, we have carefully addressed all of your comments at our best. In the following, your comments are quoted and followed by our responses.
>
> **I think a few new paper are missing, for example Counterfactual Fairness: Unidentification, Bound and Algorithm from Wu et al 2019 that bounded the reachable counterfactual fairness. It would have been interesting to see how the results related to these bounds.**
>
> Re: Thanks for pointing out this interesting paper. If we understand correctly, the counterfactual quantity in our paper is identifiable under our assumptions. Therefore we can design algorithms that could achieve Kusner's Counterfactual Fairness instead of Wu's $\tau$-Counterfactual Fairness. It would be an interesting future research topic to weaken our assumptions and see if we can design preprocessing procedures that help achieve $\tau$-Counterfactual Fairness in unidentifiable cases. Following your suggestion, we've added Wu et al. (2019) and a few other related papers as our references in this revision.
>
>
> **"I think that the statement: 'We prove that CF is equivalent to the conditional independence of the decision and the sensitive attributes given the processed non-sensitive attributes' is a bit misleading. As far as I understand, this is only true given that either condition 1 is satisfied or conditions 2 and 3 are satisfied. There exists statistical test for conditional independence but checking for condition 1, 2, 3 seems more complicated."**
>
> Re: Thank you for pointing this out. We agree that the non-sensitive conditions 1, 2 may be difficult to check in practice. It generally requires to model the dependence of non-sensitive attributes on sensitive attributes and unobserved exogenous variables. It is interesting to develop a diagnosis tool to check these conditions based on a flexible model, but this is out of the scope of the current paper and will be investigated in our future work. The theorem itself was clearly stated and the statement in the contribution section has been edited to "... CF is equivalent to the conditional independence ... under certain conditions."
>
> **"On the real data, I would like to see a discussion on the meaning of CF-metric and FLAP method given that the condition 1, 2,3 are not verified."**
>
> Re: Thank you for the suggestion. The CF-metric will be a good measure of counterfactual fairness when condition 2 is met, meaning that a counterfactually fair decision would have CF-metric equal to zero. When condition 2 is not met, there's no such guarantee, but we would expect fairer decisions to have smaller CF-metric in practice. Likewise, the FLAP method is not guaranteed to yield CF decisions but we expect it to yield fairer decisions than methods without the data preprocessing step. A discussion has been added under the definition of the CF-metric.
>
> There might be fancier or more complicated methods that could learn the counterfactual decisions under weaker conditions, but we aim to make it easy for decision-makers to use algorithms to address the fairness issue. Our method is easy to follow and can be combined with all kinds of learning methods. Thus we think it will be a good start for decision-makers to adopt and get used to the FLAP method, and we can continue to improve the preprocessing procedure and weaken the assumptions.

---

### Official Review · AnonReviewer4 · 2020-10-28
**Review of Counterfactual Fairness through Data Preprocessing**

**Rating:** 5
**Confidence:** 4

**Review:**

The paper addresses the problem of preprocessing the data in a way that the predictions of a learning task will be counterfactually fair. The counterfactual fairness definition is borrowed from that of (Kusner et al., 2017). The authors propose ortogonaliza tion and marginal distribution mapping so as to achieve counterfactual fairness. They test their proposed approach on synthetic and real data.

The math of Definition 1, Counterfactual Fairness of (Kusner et al., 2017) is wrongly stated. Counterfactual fairness is defined over the predicted $\hat{Y}$. I suppose (although am not sure) that it’s a typo because it’s correct in the proof of the theorem in the appendix.

The MAE of FLAP-1 and FLAP-2 are higher than other methods in Figure 2, although are lower in the loan application experiment. The authors can elaborate on why this is the case and provide some discussion. This is also the trend in examples 2 and 3 in Figure 4. The authors discuss the DF-metric which has shown promising results in these figures. More discussion on the MAE would be appreciated.

I would encourage the authors to run more experiments with more real-world datasets (possibly ones that are publicly available) so that the readers can get a more comprehensive comparison of the methods.

It is not clear what the difference is between the FTU predictor and the AML predictor? The text says, in Section 3.1., that “A Fairness-Through Unawareness (FTU) predictor $f_{FTU}(a)$ only uses the non-sensitive attributes” and that “An Averaged Machine Learning (AML) predictor [the mathematical expression] only needs non-sensitive attributes a as its input”. These descriptions are confusing.

---

> ### Author Response · Authors · 2020-11-22
> **Response to Reviewer4**
>
> We sincerely thank you for the encouraging and constructive comments. In this revision, we have carefully addressed all of your comments at our best. In the following, your comments are quoted and followed by our responses.
>
> **"The math of Definition 1, Counterfactual Fairness of (Kusner et al., 2017) is wrongly stated. Counterfactual fairness is defined over the predicted $\hat{Y}$. I suppose (although am not sure) that it's a typo because it's correct in the proof of the theorem in the appendix."**
>
> Re: Thank you for pointing this out. Our notation is confusing indeed, but we meant that the counterfactual fairness can be defined on any decisions that only depends on the endogenous variables A and S and an independent exogenous variable. Simply speaking, Definition 1 applies to not only the predicted decision $\hat{Y}$ but also the original decision $Y$. Most literature only discusses how to learn fair decisions and thus defining CF on $\hat{Y}$ is sufficient. Since we also care about detecting unfairness in the original decisions, we need to extend the definition of CF to $Y$. Luckily, the definition is proper for $Y$ without any modification due to the symmetry of $Y$ and $\hat{Y}$ in the structural causal model. We have edited the notation in Definition 1 to make this clear.
>
> **"The MAE of FLAP-1 and FLAP-2 are higher than other methods in Figure 2, although are lower in the loan application experiment. The authors can elaborate on why this is the case and provide some discussion. This is also the trend in examples 2 and 3 in Figure 4. The authors discuss the DF-metric which has shown promising results in these figures. More discussion on the MAE would be appreciated."**
>
> Re: Thanks for the observation and suggestion. In simulation settings where all attributes used to generate the decision $Y$ are available for predicting $\hat{Y}$, we expect fairer decisions to have higher MAE because they correct the bias of and thus deviate from the original decisions. However, this may not be the case in real-world examples where we don't have access to all the variables possessed by the original decision maker. When the original decisions depend on additional information, the FLAP methods may yield predictions closer to or farther from the original decisions, and thus the MAE on the test dataset could be lower or higher than other methods learned using the limited information. Following your suggestion, a discussion of this problem has been added to the end of Section 5.
>
> **"I would encourage the authors to run more experiments with more real-world datasets (possibly ones that are publicly available) so that the readers can get a more comprehensive comparison of the methods."**
>
> Re: Thanks for the advice. Following your suggestion, we've run two more experiments using the adult income data from UCI and the COMPAS recidivism data from ProPublica to compare the FLAP with other fairness learning methods. The results are similar and included in the appendix.
>
> **"It is not clear what the difference is between the FTU predictor and the AML predictor? The text says, in Section 3.1., that "A Fairness-Through Unawareness (FTU) predictor $f_{FTU}(a)$ only uses the non-sensitive attributes" and that "An Averaged Machine Learning (AML) predictor [the mathematical expression] only needs non-sensitive attributes a as its input". These descriptions are confusing."**
>
> Re: Sorry for the confusion. The statement about the FTU predictor is correct. For example, if we fit a logistic regression model of Y on A, then the fitted model is an FTU learner. For the AML predictor, what we meant by "only needs non-sensitive attributes $a$ as its input" is that the trained predictor will first predict one decision for the given $a$ and each possible $s\in\mathcal{S}$ and then average those predictions to give its final prediction. In this sense, we don't need to input $s$ for prediction since $\mathcal{S}$ is known.
>
> FTU and AML are similar as they both only need the non-sensitive information $a$ as the input when making the prediction. They are different at training, while FTU only uses $a$ to train the model, AML will need both $a$ and $s$ to train the machine learning model. We've rephrased the statements and removed the confusion part in this revision.

---

### Official Review · AnonReviewer3 · 2020-10-30
**A paper on an important topics, but need improvements on execution**

**Rating:** 4
**Confidence:** 2

**Review:**

This paper proposed a method to preprocess the dataset, so a machine learning classifier learned on the pre-processed data would be counterfactually fair.

Strong points:
+ The studied fairness problem is very important, and the causality-base fairness is a very challenging question
+ The proposed method is very intuitive and easy to follow

Points for Improvements:
- On the experiments part, I was wondering why the paper didn't compare to the original Counterfactual fairness paper's algorithms by Kusner et al. 2017. They have proposed three levels of algorithms to study the counterfactual fairness.
- On the experiment evaluation part, I was wondering why the paper chose MAE as the performance metrics, because we are talking about classification problems here. Are the ground truth scores just 1, and 0, and we are comparing our prediction probability to these 1 and 0's?
- Sorry if I missed the point, but I would appreciate the authors kind clarification: the dataset is preprocessed using a function we proposed, and the counterfactual metric (CF-metric) is also evaluated using this exact preprocessing function, while all other methods have no access to this function. Would this appear to be unfair to other baselines? What if say the EO, AA algorithm was also trained on the pre-processed dataset?
- The paper assumes a causal model as in Fig. 1. Does this model always hold in practice? If it is not, what should we do?
- Minor: Page 4 Section 3, Line 1, $\mathcal{S} \times \mathcal{A} \rightarrow \mathcal{A}'$, missing a '
- Minor: Above Page 3 Definition 1., U is used as unit and also previously as noise variable. Would probably be less confusing to choose another character.

---

> ### Author Response · Authors · 2020-11-22
> **Response to Reviewer3**
>
> We sincerely thank you for your encouraging and constructive comments. In this revision, we have carefully addressed all of your comments at our best. In the following, your comments are quoted and followed by our responses.
>
> **"On the experiments part, I was wondering why the paper didn't compare to the original Counterfactual fairness paper's algorithms by Kusner et al. 2017. They have proposed three levels of algorithms to study the counterfactual fairness."**
>
> Re: Thanks for the question. We didn't include the FairLearning method by Kusner et al. 2017 because Wang et al. 2019 claimed their AA algorithm can outperform it in the accuracy sense and the two methods are the same in the fairness sense as measured by the AA-metric. Follow your suggestion, we implemented FairLearning in this revision and find out that FairLearning is fairer than AA algorithm as measured by our CF-metric. However, our FLAP method is still better than FairLearning as measured by CF-metric. Please see the related results in Section 4.1.
>
> **"On the experiment evaluation part, I was wondering why the paper chose MAE as the performance metrics, because we are talking about classification problems here. Are the ground truth scores just 1, and 0, and we are comparing our prediction probability to these 1 and 0's?"**
>
> Re: Thank you for the question. Note that when we have binary outcomes and probability predictions, MAE is equal to 1 - E(accuracy), and thus it can serve the purpose of measuring the accuracy of the prediction. In this revision, we added two more metrics, the area under the ROC curve and average precision, to measure the classification performance in the appendix. It can be seen that these performance metrics basically tell the same story about the prediction/classification performance. Please see the related results in Appendix B.
>
> **"Sorry if I missed the point, but I would appreciate the authors kind clarification: the dataset is preprocessed using a function we proposed, and the counterfactual metric (CF-metric) is also evaluated using this exact preprocessing function, while all other methods have no access to this function. Would this appear to be unfair to other baselines? What if say the EO, AA algorithm was also trained on the pre-processed dataset?"**
>
> Re: Thanks for the question. First, it was a fair comparison since all methods are measured using the same metrics. Second, we understand your concern and we agree that a better metric should be completely independent of the learning algorithm, but it is still a hard open problem to find such a metric for counterfactual fairness. For example, the AA-metric defined in Wang et al.(2019) also uses a function from their AA algorithm to find the counterfactuals of the non-sensitive attributes. In fact, the fairness learning method and the fairness evaluation metric are so closely related that if we were able to find a better metric of counterfactual fairness, we could then easily design a method to achieve counterfactual fairness under that metric. In this sense, the FLAP method is easy to follow and general enough so that we could easily improve it by plugging in new preprocessing procedures when there are better metrics available. Third, when compared to the available AA metric, our CF-metric is more general because it allows us to compare more than two sensitive categories and it needs weaker conditions to be a good representative of counterfactual fairness, meaning that a counterfactually fair decision will have CF-metric equal to zero but AA-metric not equal to zero when condition 2 is met but condition 1 is not.
>
> The AA algorithm uses a similar idea of data processing to adjust the already learned decision rule and it would be redundant to use both data preprocessing and AA algorithm at the same time. That is, in situations where the AA algorithm is supposed to work (condition 1), training it with preprocessed data would not make a difference, and both AA and our algorithms would perform very similar; while in situations where the AA algorithm does not work (condition 2 holds but condition 1 does not), training AA with preprocessed data would also yield CF decisions but it is not efficient to do so as FLAP alone is sufficient to achieve CF.

---

> > ### Author Response · Authors · 2020-11-22
> > **Response to Reviewer3 (cont'd)**
> >
> > **"The paper assumes a causal model as in Fig. 1. Does this model always hold in practice? If it is not, what should we do?"**
> >
> > Re: The model may not always hold in practice. For example, the most common violation of the structural causal model in practice is the existence of unmeasured confounders, which means that $U_A$ and $U_S$ are not independent. Unfortunately, such violations are often untestable in general unless we only consider a restricted class of models. As Kusner et al. (2017) put it, "such models should be deemed provisional and prone to modifications if, for example, new data containing measurement of variables previously hidden contradict the current model." There is also literature available on the sensitivity analysis of violations of the structural causal model, such as "The sensitivity of counterfactual fairness to unmeasured confounding" by Kilbertus et al. (2020) but such analysis is beyond the scope of this paper due to the page limit. A discussion of this problem has been added to Section 6.
> >
> > **"Page 4 Section 3, Line 1, $\mathcal{S}\times\mathcal{A}\to\mathcal{A}'$, missing a '"**
> >
> > Re: Thank you for pointing this out, we've made the correction in this revision.
> >
> > **"Above Page 3 Definition 1., $U$ is used as unit and also previously as noise variable. Would probably be less confusing to choose another character."**
> >
> > Re: Sorry for the confusion. We will not refer to U as a unit in this revision. As we pointed out in the last few lines of the second paragraph in Section 2, the subset $\\{U_S, U_A, U_Y\\}$ characterizes everything we know about a unit under the structural causal model, and thus we used $U$ to index the units. After the revision, the counterfactual quantity $Y_{s}(U)$ is interpreted as the result for a random unit with exogenous variable $U$ sampled from the population had $S$ been $s$.

---

### Decision · Program_Chairs · 2021-01-07
**Final Decision**

**Decision:**

Reject

**Comment:**

The paper introduces an approach to counterfactual fairness based on data pre-processing, and compare it to other two counterfactual fairness approaches on the Adult and COMPAS datasets.

The reviewers are in agreement that, in its current state, the paper should not be accepted for publication at the venue. Their main concerns are around the metric used to measure fairness, and these were not resolved during the discussion. The reviewers would have also appreciated more experiments on real-world datasets to get a more comprehensive comparison of the methods. Finally,  discussion and comparison with other methods to achieve counterfactual fairness from the literature were limited.